# Prevalence and associated factors of skin disease among school-age children of rural district of eastern zone of Tigray, Ethiopia, 2024

Mulugeta Abraha Gebregiorgis[1][¤a]*, Almaz Berhe[2][¤b], Haftom Gebrehiwot[2][¤b], Mamush Gidey Abrha[1][¤a], Binyam Gebrehiwet Tesfay[1][¤a], Fiseha Abadi Gebreanenia[3][¤c], Binyam Tsegay Hagos[1][¤a], Willi Bahre[1][¤a], Guesh Teklu Woldemariam[1][¤a], Mearg Alemu Halefom[1][¤a], Tesfay Gebreslassie Gebrehiwot[1][¤a]

1 College of Medicine and Health Sciences, Adigrat University, Adigrat, Tigray, Ethiopia, 2 College of Health Science, School of Nursing, Mekelle University, Mekelle, Tigray, Ethiopia, 3 College of Health Sciences, Axum University, Axum, Tigray, Ethiopia

¤a current address: Adigrat, Northern Ethiopia
¤b current address: Mekelle, Northern Ethiopia
¤c current address: Axum, Northern Ethiopia

* mullerab22@gmail.com

## Abstract

### Background

Skin diseases are conditions that affect the skin. Skin diseases encompass various conditions, including infections, allergies, inflammatory disorders, and infestation. Skin diseases are a significant public health concern, particularly among school-age children, as they can cause discomfort, and social stigma, and influence overall well-being. There is limited research conducted specifically among school-age children, especially in the postwar period, hindering the understanding of the extent of the problem and the identification of potential contributing factors. In Ethiopia, a study conducted on primary schoolchildren shows that the prevalence of skin diseases was 61.2%.

### Methods

A school-based cross-sectional study was employed from September 10 to October 30, 2024, in selected schools of Eastern Tigray. A multi-stage sampling technique was used to choose Woredas and schools. Subsequently, 603 school-age children were recruited using systematic random sampling. Data were collected through interviewer-administered structured questionnaires and physical examination. Binary logistic regression was used to determine the strength of the association. Variables with a p-value <0.2 were selected for multivariable analysis, and a p-value <0.05 in the final model was considered statistically significant.

**Data availability statement:** All relevant data are within the manuscript.

**Funding:** The author(s) received no specific funding for this work.

**Competing interests:** The authors have declared that no competing interests exist.

**Abbreviations:** CI, Confidence Interval; DALY, Disability Adjusted Life Years; IDP, Internally Displaced Person; NGOs, Non-Governmental Organizations; OR, Odds Ratio; PIH, Post Inflammatory Hyperpigmentation; SAC, School-Age Children; SD, Skin Disease; SPSS, Statistical Package for Social Science; WHO, World Health Organization; YLDs, Year Lost due to Disability

## Result

The prevalence of skin disease among school-age children in the Rural District of the Eastern Zone was 411 (68.2%). Moreover, the factors associated with skin disease in this study were living in IDP (AOR = 2.96; 95% CI: 1.86–4.7), families with history of skin disease (AOR = 8.22; 95% CI: 5.19–13), sharing personal items (AOR = 3.89; 95% CI: 2.11–7.17), children who sometimes wash their hands before meals (AOR = 1.97; 95% CI: 0.99–3.89), and children who never wash their hands regularly (AOR = 4.93; 95% CI: 2.66–9.11).

## Conclusion

The prevalence of skin disease among school-age children was high. Proper hand hygiene, not sharing personal items, and treating family members with skin diseases can help decrease the prevalence of skin diseases.

### Introduction

Skin diseases are conditions that affect the skin. These diseases may cause rashes, inflammation, itchiness or other skin changes. Some skin conditions may be genetic, while lifestyle factors may cause others. Skin disease treatment may include medications, creams, ointments, or lifestyle changes [1,2].

Skin conditions incorporate various disorders, including infectious diseases, allergic reactions, inflammatory problems, and infestations [1]. These skin issues can manifest with varying degrees of severity, from minor irritations to chronic and disabling conditions. School-age children are especially susceptible to developing skin diseases due to their maturing immune systems, close interactions within the school setting, and participation in outdoor activities [1,2].

Skin and subcutaneous diseases were responsible for 41.6 million Disability Adjusted Life Years (DALYs) and 39.0 million years lost due to Disability (YLDs) which makes them the 18th leading cause of global DALYs. Excluding mortality, skin diseases were the fourth leading cause of disability worldwide [3].

Dermatological conditions are a common problem in school-age children. Contact between classmates is an important cause of skin infections and infestations. The occurrence and pattern of skin diseases have been seen to vary depending on the socio-economic and cultural factors related to hygiene and treatment-seeking behavior. Skin diseases cause discomfort, and social stigma, and impact overall well-being [4,5].

Dermal issues are one of the most important health matters with significant impacts on school-aged children and cause nonfatal disabilities worldwide, particularly in low-resource regions [6].

Skin-related ailments affect people of all ages, yet children are one of the most common victims of it [3]. Skin diseases represent an important part of the morbidity among children and are possibly influenced by, social, and economic factors [7].

Understanding the prevalence and potential factors associated with the occurrence of skin diseases is crucial for effective prevention and management strategies [8]. The most epidemiologically important seems to be the infectious types because of their transmissibility and amenability to simple school-health measures [9].

It is increasingly recognized that schools play an important role in coaching healthy and hygienic habits among the younger generations, by take-home messages, which create awareness among the children, parents, or guardians on healthy living. [4].

The Rural District of Eastern Zone in the Tigray Region of Ethiopia has distinct environmental, cultural, and socioeconomic factors that may affect the prevalence and risk factors associated with skin diseases among school-age children in the area. The Tigray Region has experienced conflict, resulting in many displaced and orphaned children in the local population. However, there is limited research that has specifically focused on this particular post-conflict population, hindering the understanding of the extent of skin disease issues and the identification of potential contributing factors in this community [10].

## Methods and materials

### Study design, population, area and period

A cross-sectional institutional study was conducted among 603 school-age children in three district Rural districts Eastern zone of Tigray, Regional State of Ethiopia. Data were collected from September 10, 2024, to October 30, 2024.

### Eligibility criteria

All school-age children between the ages of 6 and 12-years children

**Inclusion Criteria.** School-age children between the ages of 6 and 12 years and children who attend primary schools within the study were included.

**Exclusion Criteria.** Children whose parents or guardians do not provide informed consent for their participation in the study were excluded.

### Sampling procedure and technique

This study utilized a multi-stage sampling approach, which involved dividing the population into clusters and randomly selecting clusters for sampling. Within each selected cluster, schools were sampled using simple random sampling. The final sample of students was then selected using systematic random sampling, where the interval (K) for selection was determined.

Initially, the primary schools providing education to school-age children in the Rural Districts of the Eastern Zone of Tigray were identified. Subsequently, the sample size necessary for each school was determined through proportional allocation. A systematic selection process was employed to choose school-age children from each institution.

### Data collection method and tools

Ten nurses and one supervisor without affiliation with the study school collected data using a structured interviewer-administered questionnaire and physical examination. The questionnaire included four main sections. The trained health professionals (nurses or health officers) clinically examined the exposed parts of the body in a well-illuminated private classroom to document the findings.

### Variables

**Dependent variables.** Skin diseases
**Independent variable.** **Socio-demographic information:** Age, Sex, Grade, Resident….

**Medical history:** Family history of previous skin disease, Diagnosis of skin disease Receipt of treatment for skin disease
**Hygiene and sanitary conditions**: Bathing frequency, Use of soap, Cloth change frequency

## Statistical analysis and management

Data were coded and entered into Epi Info version 7.2 and cross-checked for consistency and accuracy. After data cleaning, it was exported to SPSS version 27 for statistical analysis. Descriptive statistics were performed, and results were summarized using tables, texts, and figures. Frequency distribution and percentages were employed for categorical variables. Binary logistic regression was used to determine the magnitude and strength of association between a set of independent variables and the outcome variable at a $p < 0.20$ significance level. Variables significant at $p < 0.20$ with the outcome variable were selected for multivariate analysis. An odds ratio with a 95% confidence level was computed, and a p-value $< 0.05$ was used as the significance level. The Hosmer-Lemeshow goodness-of-fit model coefficients test procedure was used to test for model fitting.

## Data quality assurance

To ensure data quality, a questionnaire was adapted from published literature, translated, and pretested. Data collectors received training and were supervised during data collection. Completed questionnaires were checked for errors. Reliability was assessed using inter-rater reliability and Cronbach's alpha, while face validity was evaluated.

## Ethical considerations

An official ethical clearance letter was obtained from the Mekelle University College of Health Sciences Institutional Review Board (IRB) on 30 July 2024 with Protocol of Approval MU-IRB 2278/2024. Moreover, the members of the IRB or ethics committee who approved this study were Professor Haftu Berhe Gebru, Dr Tesfay Gebregzabher Gebrehiwot, Dr Mengistu Welday Gebremihael, Dr Meskelu Kidu Weldetensae, Dr Abraha Hailu Weldegerima, Dr Gebretsadik Berhe, and Gidey Gebremeskel. A formal letter for permission and support was written to the respective administrator's office. The purpose of the study was clearly explained, informed oral consent was maintained, confidentiality was ensured, and assent was obtained from the study subjects who are greater than eight years old. Parents were given consent forms for their written approval before data collection began. Detailed information about the child was subsequently gathered from their parents. Parents who were unwilling to participate from the beginning or any part of the interview were allowed to do so. There was no risk of danger or hazardous procedures putting the participants in harm.

## Result

In this study, a total of 603 out of 603 samples were included, yielding a response rate of 100%.

### Socio demographic characteristics of study participants

The response rate was 100% (603), among them, 289 (47.9%) were male. The mean age of the participants was 9.41 years (±1.66 SD), and the average number of rooms in their households was 1.51 (±1.04). The total number of beds in the school-aged children's home ranged from a minimum of 1 to a maximum of 4. Out of 603 participants, 416 (69%) lived with their mother and father. Additionally, the educational status of caregivers, with the category of illiterate, accounted for 459 (76.1%). Furthermore, 208 (65.5%) of the participants were from the host community, and 483 (80.1%) caregivers identified as farmers (Table 1).

### Medical History

From the total of 603 participants, 212 (35.2%) caregivers had a previous history of skin disease. Among these caregivers, 65 (10.8%) had skin disease for less than one month, while 89 (14.8%) caregivers had skin disease for a duration of 1–6

**Table 1. Socio-demographic status of study participants in the prevalence of skin disease and associated factors among school-age children of Rural District of Eastern Zone, Northern Ethiopia, 2024.**

| Variables | Response | Frequency | Percent |
|---|---|---|---|
| Sex | Male | 289 | 47.9 |
| | Female | 314 | 52.1 |
| Child lives with | Parent | 416 | 69.0 |
| | Mother only | 119 | 19.7 |
| | Father only | 25 | 4.10 |
| | Grandparent | 24 | 4.00 |
| | Other | 16 | 3.20 |
| Mothers/Father/caregiver's educational status | Illiterate | 459 | 76.1 |
| | Read and write | 98 | 16.3 |
| | Elementary | 29 | 4.80 |
| | High school | 17 | 2.80 |
| Mother/ Father/ caregiver's Occupation | Jobless | 54 | 9.00 |
| | Merchant | 18 | 3.00 |
| | Government employed | 28 | 4.60 |
| | Farmer | 483 | 80.1 |
| | Private employed | 20 | 3.30 |
| Community Status | Host | 395 | 65.5 |
| | IDP | 208 | 34.5 |

months, and 46 (7.6%) caregivers noted that the condition had persisted for more than one year. Of the 168(28%) family members diagnosed with a skin disease, 167 (27.7%) received treatment. Finally, 20 (3.3%) of the children had underlying health conditions. (Table 2).

## Hygiene and sanitary conditions

Among 603 participants, 519 (86.1%) take a bath or shower fewer than four times a month. The use of soap during bathing was reported by 595 (98.7%) individuals. From the total of 603 participants, 4 (6.6%) distinguished that they never shared personal items, while 60 (10.0%) indicated they always did. Only 187 (31.0%) participants wash their hands after using the restroom, and 84 (13.9%) wash their hands before meals, with the remaining 144 (23.9%) participants not washing their hands regularly. Regarding the frequency of changing clothes, 162 (26.9%) changed clothes daily, while 285 (47.3%) changed them once a week. Access to clean water for bathing and washing was available to 568 (94.2%) participants. Lastly, only 17 (2.8%) reported using tap water, while 80 (13.3%) stated they used river water (Table 3).

## Prevalence of skin disease

The prevalence of skin disease among the school-aged children in Rural Districts of the Eastern Zone of Tigray was 411 (68.2%) (95% CI: 64.2–72.0).

## The common types of skin diseases identified

The frequency of various types of skin diseases reveals a diverse spectrum of conditions. Based on the three categories of skin disease specified in this study, infections account for 36%, allergic dermatitis for 12.8% and trauma related conditions for 18.7% (Fig 1).

Variables with p value less than 0.2 in the bivariate analysis were included in the multivariable logistic regression. These variables were: community living residence status (IDP or Host), living arrangements (with mother, father, or

**Table 2. Medical History of Participants in the Prevalence of Skin Disease and Associated Factors among School-age Children of Rural District of Eastern Zone, Northern Ethiopia, 2024.**

| Variables | Category | Frequency | Percent |
|---|---|---|---|
| Positive Family History of skin disease | Yes | 212 | 35.2 |
| Have you ever been diagnosed with a skin disease? | Yes | 168 | 28.0 |
| Have you received treatment for your skin disease(s)? | Yes | 167 | 27.7 |
| Duration of the skin disease | Less 1 month | 65 | 10.8 |
| | 1-6 month | 89 | 14.9 |
| | 6-12 month | 12 | 1.90 |
| | More than 1 year | 46 | 7.60 |
| Do you have any underlying health conditions? | Yes | 20 | 3.30 |
| | No | 583 | 96.7 |
| If yes, please specify the condition(s) | Diabetes Mellitus (DM) | 4 | 0.70 |
| | HIV | 1 | 0.20 |
| | Autoimmune condition | 10 | 1.70 |
| | Epilepsy | 3 | 0.50 |
| | Other | 2 | 0.20 |

**Table 3. Hygiene and sanitary conditions of participants in the prevalence of skin disease and associated factors among school-age children of Rural District of Eastern Zone, Northern Ethiopia, 2024.**

| Variables | Category | Frequency | Percent |
|---|---|---|---|
| How often do you bathe or shower? | A few times a week | 36 | 6.00 |
| | Once a week | 48 | 8.00 |
| | Less than 4 times per month | 519 | 86.1 |
| Do you use soap when bathing or showering? | Yes | 595 | 98.7 |
| Frequency of close sharing: | Never | 4 | 6.60 |
| | Rarely | 123 | 20.4 |
| | Sometimes | 217 | 36.0 |
| | Often | 163 | 27.0 |
| | Always | 60 | 10.0 |
| Hand washing practices: | Always wash hands after using the restroom | 187 | 31.0 |
| | Always wash hands before meals | 84 | 13.9 |
| | Sometimes wash hands after using the restroom | 178 | 12.9 |
| | Sometimes wash hands before meals | 110 | 18.2 |
| | Never wash hands regularly | 144 | 23.9 |

caregiver), caregiver's educational status and occupation, family history of skin disease, frequency of bathing, cloth sharing habits, hand washing practices, frequency of cloth changing, sharing of personal items, presence of domestic animals, and type of water used (Table 4).

## Factors associated with skin diseases among school-aged children

According to the multivariate logistic regression analysis, four (Community status, Family history of skin disease, Handwashing practices, and sharing of personal items) variables were significantly associated with the occurrence of skin disease. Therefore, the odds of having skin disease were 2.96 (95% CI: 1.86–4.7) higher among the school-aged children living in IDP compared to the school aged children living in the host community. The odds of having skin disease is 8.22(95%

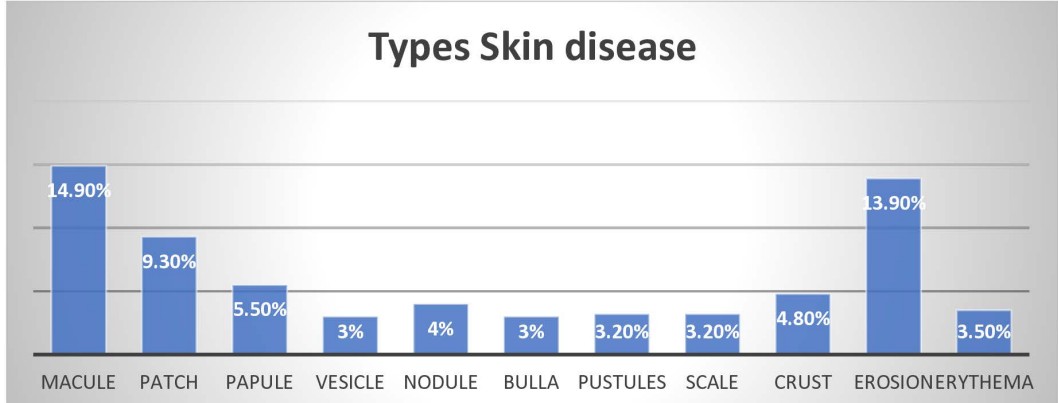

**Fig 1. Types of skin disease among school-aged children of rural district of Eastern Zone of Tigray Ethioipia 2024.**

CI: 5.19–13) higher among families with a history of skin disease compared to families without a history of skin disease. The odds of having skin disease is 3.89 (95% CI: 2.11–7.17) higher among children who share personal items compared to children who did not share personal items. The odds of having skin disease is 4.93 (95% CI: 2.66–9.11) higher among those who never wash their hands regularly compared to those who always wash their hands after using the toilet. (Table 5).

## Discussion

The primary objective of this study is to assess the prevalence and factors associated with the occurrence of skin diseases among school-age children in the Rural District of the Eastern Zone of Tigray, Ethiopia. Understanding the prevalence and associated factors is vital for developing targeted interventions to mitigate skin diseases in this vulnerable population. The findings of this study revealed a notable prevalence of skin diseases among school-age children, recorded at 68.2%. This high prevalence underscores the urgent need for public health initiatives aimed at improving skin health in rural communities. Additionally, several factors were identified as significantly associated with the occurrence of skin diseases, including community status as Internally Displaced Persons (IDP), a previous family history of skin disease, hand-washing practices, and the sharing of personal equipment. Each of these factors highlights critical areas for intervention and further investigation to reduce the burden of skin diseases in this population.

The findings revealed that the prevalence of skin disease among school-age children was 68.2%. Moreover, community residence status as IDP, previous family history of skin disease, sharing of personal items, and never washing their hands regularly were significantly associated variables in this study. However, the occupation of the parent as a farmer was not associated, as the high proportion of farmers may dilute the impact of occupation on skin health, which could be because over 80% of the parents were farmers.

The findings of this study have a higher prevalence of skin disease than the findings in Sri Lanka (63.9%) [4], India (60.59%) [11], Southern Co◦te d'Ivoire (25.6%) [12], Colombia (42.8%) [7], Ethiopia (61.2%) [3]. The possible explanation for this high prevalence may be due to the conflict in Tigray, which damaged hygiene and sanitary facilities in schools and communities. Moreover, due to the conflict, people were displaced from their residences and became vulnerable to skin diseases due to a lack of access to clean water, soap, latrines, and even shelter. All of these factors may contribute to the higher prevalence of skin diseases among school-age children. This explanation is supported by a study on water supply infrastructure [13].

The hand hygiene practice was also found to increase the odds of never washing hands regularly compared to those who washed their hands regularly, with an Adjusted Odds Ratio of 4.93. This finding was higher than a study conducted

**Table 4. Bivariate analysis of factors associated with skin disease among school-age children of Rural District of Eastern Zone, Northern Ethiopia, 2024.**

| Variable | Category | Skin disease | | COR (95%CI) | P-value |
|---|---|---|---|---|---|
| | | Yes | No | | |
| Community status (IDP or Host) | Host | 297 | 98 | 1 | |
| | IDP | 114 | 94 | 2.49 (1.75-3.57) | <.001 |
| Child lives with | Parents | 300 | 116 | 1 | |
| | Mother Only | 76 | 43 | 1.46 (0.95-2.25) | 0.083 |
| | Father Only | 11 | 14 | 3.29 (1.45-7.46) | 0.004 |
| | Grand Parent | 14 | 10 | 1.85 (0.79-4.28) | 0.152 |
| | Other | 10 | 9 | 2.33(0.92-5.87) | 0.074 |
| Mothers/Father/caregiver's educational status | 1. Illiterate | 322 | 137 | 0.48 (0.18-1.27) | 0.140 |
| | 2. Read and write | 60 | 38 | 0.71 (0.25-2.00) | 0.521 |
| | Elementary education | 20 | 9 | 0.50(0.15-1.74) | 0.280 |
| | High School Education | 9 | 8 | 1 | |
| Mothers/ Father/ caregiver's Occupation | 3. Jobless | 42 | 12 | 0.28(0.09-0.847) | 0.024 |
| | 4. Merchant | 8 | 10 | 1.25(0.35-4.49) | 0.732 |
| | 5. Farmer | 338 | 145 | 0.43(0.17-1.05) | 0.065 |
| | Government employed | 13 | 15 | 1.15(0.37-3.6) | 0.807 |
| | Private employed | 10 | 10 | 1 | |
| Family history of skin disease | Yes | 89 | 123 | 6.45(4.25-9.40) | <.001 |
| | No | 322 | 69 | 1 | |
| How often do you bathe or shower? | A few times a week | 18 | 18 | 1 | |
| | Once a week | 34 | 14 | 0.41(0.16-1.01) | 0.054 |
| | Less than 4 times per month | 359 | 160 | 0.44(0.22-0.88) | 0.020 |
| Frequency of close sharing: | • Never | 24 | 16 | 1 | |
| | • Rarely | 82 | 41 | 0.75(0.36-1.56) | 0.443 |
| | • Sometimes | 137 | 80 | 0.87(0.43-1.75) | 0.707 |
| | Often | 119 | 44 | 0.55(0.27-1.14) | 0.109 |
| | Always | 49 | 11 | 0.34(0.14-0.84) | 0.019 |
| Hand washing practices: | Always wash hands after using the restroom | 147 | 40 | 1 | |
| | Always wash hands before meals | 60 | 24 | 1.47(0.82-2.65) | 0.199 |
| | Sometimes wash hands after using the restroom | 61 | 17 | 1.02(0.54-1.94) | 0.942 |
| | Sometimes wash hands before meals | 75 | 35 | 1.71(1.00-2.92) | 0.047 |
| | Never wash hands regularly | 68 | 76 | 4.10(2.54-6.63) | <.001 |
| How often do you change your clothes? | Daily | 118 | 44 | 0.82(0.53-1.25) | 0.366 |
| | Few times a week | 97 | 59 | 1.34(0.89-2.01) | 0.162 |
| | Once a week and above | 196 | 89 | 1 | |
| Do you share personal items (towels, clothes, etc.) | Yes | 35 | 56 | 4.42(2.78-7.04) | <.001 |
| | No | 376 | 136 | 1 | |
| Presence of domestic animal in the house | Yes | 398 | 181 | 0.53(0.23-1.22) | 0.139 |
| | No | 13 | 11 | 1 | |
| What type of water do you use? | Tap water | 8 | 9 | 1 | |
| | River | 59 | 21 | 0.32(0.10-0.93) | 0.036 |
| | Wells | 344 | 162 | 0.42(0.16-0.16) | 0.080 |

**Table 5. Multivariate analysis of factors associated with skin disease among school-age children of rural district of eastern zone, northern Ethiopia, 2024.**

| Variable | Category | Skin disease | | AOR (95%CI) | p-value |
|---|---|---|---|---|---|
| | | Yes | No | | |
| Community status (IDP or Host) | Host | 297 | 98 | 1 | |
| | IDP | 114 | 94 | 2.96(1.86-4.7) | <0.001 |
| Family history of skin disease | Yes | 89 | 123 | 8.22(5.19-13) | <0.001 |
| | No | 322 | 69 | 1 | |
| Hand-washing practices: | Always wash hands after using the restroom | 147 | 40 | 1 | |
| | Always wash hands before meals | 60 | 24 | 1.09(0.51-2.3) | <0.819 |
| | Sometimes wash hands after using the restroom | 61 | 17 | 0.97(0.43-2.23) | <0.957 |
| | Sometimes wash hands before meals | 75 | 35 | 1.97(0.99-3.89) | <0.052 |
| | Never wash hands regularly | 68 | 76 | 4.93(2.66-9.11) | <0.001 |
| Do you share personal items (towels, clothes, etc.) | Yes | 35 | 56 | 3.89(2.11-7.17) | <0.001 |
| | No | 376 | 136 | 1 | |

in Northern Ethiopia, where the Adjusted Odds Ratio was 1.78. [1]. The possible reason could be Limited access to clean water, soap, and proper hand-washing facilities can significantly affect hand hygiene practices in rural communities. Additionally, a lack of educational programs promoting the importance of hand hygiene may result in low awareness and poor practices among children and families. Without both the necessary resources and the knowledge of their importance, maintaining consistent hand washing habits becomes challenging [14,15].

This study found that sharing personal items, such as clothes and towels, significantly increased the odds of having skin diseases compared to those who did not share any personal items, with an adjusted Odds Ratio of 3.89. This finding was higher than a study conducted in Northern Ethiopia (Debre Berhan town) where the Odds Ratio was 1.5 [3]. The possible reason for the high Odds of skin diseases due to sharing personal items could be a lack of clothes and other personal items like towels. This might be because this study was conducted among school-age children living in rural areas, and the effects of post-war conditions [16].

This study confirmed that participants with a family history of skin disease were at an increased Odds of skin disease compared to their counterparts, with an adjusted odds ratio of 8.22. The possible explanation might be due to the war in Tigray damaged health services and the living conditions of the community such as sleeping in one room [1].

According to this study, the Odds of having skin disease were significantly increased among IDPs as compared with the host community, with an Adjusted Odds Ratio of 2.96. This finding was in line with the study conducted in Armachiho district, 3.47 [17]. The possible explanation is that both studies included internally displaced People, who are vulnerable to skin disease due to a lack of access to clean water, detergents, Living conditions, Economic status, hygiene, and sanitary facilities [18].

The study highlights a high prevalence of skin diseases among school-age children in the Rural District of the Eastern Zone of Tigray, emphasizing the urgent need for public health interventions. Contributing factors include community status as Internally Displaced Persons, family history of skin disease, poor hand hygiene, and sharing personal items. The impact of conflict has disrupted sanitation and access to clean water, exacerbating these issues. These findings underscore the necessity for targeted strategies to improve skin health in this vulnerable population.

## Strength and limitations of the study

### Strength of the study.

• The cross-sectional design of the study prevented the determination of causal relationships.

- The study was conducted over a limited timeframe, which may have overlooked seasonal variations in skin diseases that could influence the findings

## Conclusion

The prevalence of skin disease among school-aged children in the Rural District of the Eastern Zone was found to be remarkably high, representing nearly two-thirds of the population. Community residence status as IDP, previous family history of skin disease, sharing of personal items, and children who never wash their hands, were significantly associated variables in this study. The prevalence of skin disease among school-age children was high. The majority of identified skin diseases were Macule and Erosion.

## Acknowledgments

I would like to express my sincere gratitude to my data collectors and supervisors for their facilitation, organization and collection of the data throughout the study period.

## Author contributions

**Conceptualization:** Mulugeta Abraha Gebregiorgis, Almaz Berhe, Haftom Gebrehiwot.

**Data curation:** Mulugeta Abraha Gebregiorgis, Binyam Gebrehiwet Tesfay, Fiseha Abadi Gebreanenia, Mamush Gidey Abrha, Guesh Teklu Woldemariam, Mearg Alemu Halefom, Binyam Tsegay Hagos, Tesfay Gebreslassie Gebrehiwot, Willi Bahre.

**Formal analysis:** Mulugeta Abraha Gebregiorgis, Almaz Berhe, Haftom Gebrehiwot, Fiseha Abadi Gebreanenia, Mamush Gidey Abrha, Mearg Alemu Halefom, Binyam Tsegay Hagos, Tesfay Gebreslassie Gebrehiwot, Willi Bahre.

**Investigation:** Mulugeta Abraha Gebregiorgis, Almaz Berhe, Haftom Gebrehiwot.

**Methodology:** Mulugeta Abraha Gebregiorgis.

**Project administration:** Mulugeta Abraha Gebregiorgis.

**Software:** Mulugeta Abraha Gebregiorgis.

**Supervision:** Almaz Berhe, Haftom Gebrehiwot.

**Writing – original draft:** Mulugeta Abraha Gebregiorgis, Almaz Berhe, Haftom Gebrehiwot.

**Writing – review & editing:** Mulugeta Abraha Gebregiorgis, Almaz Berhe, Haftom Gebrehiwot, Binyam Gebrehiwet Tesfay, Guesh Teklu Woldemariam.

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
