## [Decision Letter · Decision Letter 0]

11 Aug 2025

Dear Dr. Gebregiorgis,

We look forward to receiving your revised manuscript.

Kind regards,

Dawit Getachew Gebeyehu, MPH

Academic Editor

PLOS ONE

Journal Requirements:

2. We note that your Data Availability Statement is currently as follows: All relevant data are within the manuscript and in Supporting Information files.

5. Please include a copy of Table 5 which you refer to in your text on page 22.

Reviewers' comments:

Reviewer's Responses to Questions

**Comments to the Author**

1. Is the manuscript technically sound, and do the data support the conclusions?

Reviewer #1: No

Reviewer #2: Yes

2. Has the statistical analysis been performed appropriately and rigorously?

Reviewer #1: N/A

Reviewer #2: Yes

3. Have the authors made all data underlying the findings in their manuscript fully available?

Reviewer #1: No

Reviewer #2: Yes

4. Is the manuscript presented in an intelligible fashion and written in standard English?

Reviewer #1: Yes

Reviewer #2: No

Reviewer #1: This study provides an overview of skin conditions among school-aged children but does not specify particular skin diseases. The findings mainly describe general skin conditions such as erosion, which may potentially result from trauma rather than a defined dermatological disorders. They also mention handwashing as associated with skin diseases, but the specific types of skin conditions are not specified. Furthermore, they did not provide detailed information on nail or hair diseases, which are typically classified as conditions. Therefore, the data on the prevalence of skin conditions is somewhat limited and lacks detail.

Although trained medical staffs, such as nurses, can perform initial physical examinations, confirmation by licensed physicians is recommended. It would also be beneficial to categorize skin diseases into specific groups, such as infections (e.g., scabies), trauma-related issues, and dermatoses like atopic dermatitis or seborrheic dermatitis, to improve clarity and diagnosis accuracy.

Additionally, the associated factors for skin conditions is limited, with most focusing on caregivers' conditions. They should also inquire about participants' birth history, vaccination status, or activities involving water or sun exposure that may be related to skin diseases.

For presenting numbers, the authors should ensure consistency by using two decimal places (e.g., 5.00) for clarity and uniformity.

They should include abbreviations under the table and also define them in the abstract when they first appear.

Reviewer #2: Please consider:

- professional English editing

- improving tables 2 & 3 as follows:

* under the column Variables: instead of "Family History of skin disease" change to: "Positive family history of skin disease", then remove the column Response, and keep the Frequency and Percentage for the answer "Yes" only.

* please do the same for all other variables.

* for variable with responses other than yes/no, move the responses and list them in the same column as the Variables

- adding the diagnoses rather than the morphology of skin diseases under the title "The common types of skin diseases identified". I think the morphology alone may not be accurate. Adding the diagnosis will help making any association between independent and dependent variables more reasonable. E.g. kids washing their hands frequently are expected to have atopic dermatitis. Kids who dont wash their hands are expected to have infections or infestations. Kids with a positive family history that is recent are expected to have infection or infestation.

- correcting the number of the table titled "Bivariate analysis of factors associated with skin disease among school-age children of Rural District of Eastern Zone, Northern Ethiopia, 2024" from 3 to 4

- Correcting the number of the table titled "Multivariate analysis of factors associated with skin disease among school-age children of rural district of eastern zone, northern Ethiopia, 2024" from 4 to 5.

- Removing figure 1, it is not necessary.

**Do you want your identity to be public for this peer review?** For information about this choice, including consent withdrawal, please see our Privacy Policy

Reviewer #1: No

Reviewer #2: No

---

## [Author Response · Author response to Decision Letter 1]

22 Jan 2026

Academic Editor comments

Answer:

We accept the comment and we made corrections according the PLOS ONE guideline.

2. We note that your Data Availability Statement is currently as follows: All relevant data are within the manuscript and in Supporting Information files.

Answer:

The submission contains all raw data

2 a. If there are ethical or legal restrictions on sharing a de-identified data set, please explain them in detail (e.g., data contain potentially sensitive information, data are owned by a third-party organization, etc.) and who has imposed them (e.g., an ethics committee). Please also provide contact information for a data access committee, ethics committee, or other institutional body to which data requests may be sent. If data are owned by a third party, please indicate how others may request data access.

Answer:

We accept the comment, and there are no ethical or legal restrictions on data access. We have already sent the data as supporting information

Answer:

I accept the comment, and I have registered in ORCID and obtained a new ID

Answer:

We accept the comment, and we include the full name of IRB in the Methods section.

5. Please include a copy of Table 5 which you refer to in your text on page 22.

Answer:

The comment is well taken and we make revision accordingly

Answer:

We accept the comment, and We make revision

Reviewer#1

1. This study provides an overview of skin conditions among school-aged children but does not specify particular skin diseases. The findings mainly describe general skin conditions such as erosion, which may potentially result from trauma rather than a defined dermatological disorders. They also mention handwashing as associated with skin diseases, but the specific types of skin conditions are not specified. Furthermore, they did not provide detailed information on nail or hair diseases, which are typically classified as conditions. Therefore, the data on the prevalence of skin conditions is somewhat limited and lacks detail. Although trained medical staffs, such as nurses, can perform initial physical examinations, confirmation by licensed physicians is recommended. It would also be beneficial to categorize skin diseases into specific groups, such as infections (e.g., scabies), trauma-related issues, and dermatoses like atopic dermatitis or seborrheic dermatitis, to improve clarity and diagnosis accuracy.

Answer:

We accept the comment and we make revision accordingly

2. Additionally, the associated factors for skin conditions is limited, with most focusing on caregivers' conditions. They should also inquire about participants' birth history, vaccination status, or activities involving water or sun exposure that may be related to skin diseases.

For presenting numbers, the authors should ensure consistency by using two decimal places (e.g., 5.00) for clarity and uniformity.

They should include abbreviations under the table and also define them in the abstract when they first appear.

Answer:

We accept the comment, and We make modification accordingly.

Reviewer #2:

1. professional English editing

• Improving tables 2 & 3 as follows:

2. under the column Variables: instead of "Family History of skin disease" change to: "Positive family history of skin disease", then remove the column Response, and keep the Frequency and Percentage for the answer "Yes" only.

3. please do the same for all other variables.

4. for variable with responses other than yes/no, move the responses and list them in the same column as the variables

Answer:

We accept the comment, and we have made changes based on the comments

6. Adding the diagnoses rather than the morphology of skin diseases under the title "The common types of skin diseases identified". I think the morphology alone may not be accurate. Adding the diagnosis will help making any association between independent and dependent variables more reasonable. E.g. kids washing their hands frequently are expected to have atopic dermatitis. Kids who don’t wash their hands are expected to have infections or infestations. Kids with a positive family history that is recent are expected to have infection or infestation.

Answer:

The comment is well taken and we make revision

6. • Correcting the number of the table titled "Bivariate analysis of factors associated with skin disease among school-age children of Rural District of Eastern Zone, Northern Ethiopia, 2024" from 3 to 4

• Correcting the number of the table titled "Multivariate analysis of factors associated with skin disease among school-age children of rural district of eastern zone, northern Ethiopia, 2024" from 4 to 5.

• Removing figure 1, it is not necessary.

Answer:

We accept the comment, and we make correction accordingly

---

## [Decision Letter · Decision Letter 1]

28 Jan 2026

Prevalence and associated factors of skin disease among school-age children of rural district of eastern zone of Tigray, Ethiopia, 2024

PONE-D-25-26829R1

Dear Dr. Gebregiorgis,

We’re pleased to inform you that your manuscript has been judged scientifically suitable for publication and will be formally accepted for publication once it meets all outstanding technical requirements.

Kind regards,

Dawit Getachew Gebeyehu, MPH

Academic Editor

PLOS One

Additional Editor Comments (optional):

Reviewers' comments:

Reviewer's Responses to Questions

**Comments to the Author**

Reviewer #2: (No Response)

2. Is the manuscript technically sound, and do the data support the conclusions?

Reviewer #2: Yes

3. Has the statistical analysis been performed appropriately and rigorously?

Reviewer #2: Yes

4. Have the authors made all data underlying the findings in their manuscript fully available?

Reviewer #2: Yes

5. Is the manuscript presented in an intelligible fashion and written in standard English?

Reviewer #2: Yes

Reviewer #2: The authors have satisfactorily addressed my comments and suggestions...............................

**Do you want your identity to be public for this peer review?** For information about this choice, including consent withdrawal, please see our Privacy Policy

Reviewer #2: No

---

## [Editor Report · Acceptance letter]

PONE-D-25-26829R1

PLOS One

Dear Dr. Gebregiorgis,

I'm pleased to inform you that your manuscript has been deemed suitable for publication in PLOS One. Congratulations! Your manuscript is now being handed over to our production team.

Kind regards,

on behalf of

Mr. Dawit Getachew Gebeyehu

Academic Editor

PLOS One